# Identification of Novel Biomarkers in Late Preterm Neonates with Respiratory Distress Syndrome (RDS) Using Urinary Metabolomic Analysis

**DOI:** 10.3390/metabo13050644

**Published:** 2023-05-09

**Authors:** Irene Christopoulou, Eirini Kostopoulou, Konstantina Matzarapi, Styliani A. Chasapi, Georgios A. Spyroulias, Anastasia Varvarigou

**Affiliations:** 1Department of Paediatrics, University of Patras Medical School, General University Hospital, 26500 Patras, Greece; i.christopoulou@shso.org.cy (I.C.); nata-var@med.upatras.gr (A.V.); 2Department of Pharmacy, University of Patras, 26504 Patras, Greece; kwnmatz95@gmail.com (K.M.); stella.chimic@gmail.com (S.A.C.); g.a.spyroulias@upatras.gr (G.A.S.)

**Keywords:** late preterm neonates, neonatal intensive care unit/NICU, urine metabolomics, metabolic profile, NMR spectroscopy, respiratory distress syndrome/RDS, nutrition, personalized medicine

## Abstract

Urine metabolomics is gaining traction as a means of identifying metabolic signatures associated with health and disease states. Thirty-one (31) late preterm (LP) neonates admitted to the neonatal intensive care unit (NICU) and 23 age-matched healthy LPs admitted to the maternity ward of a tertiary hospital were included in the study. Proton nuclear magnetic resonance (^1^H NMR) spectroscopy was employed for urine metabolomic analysis on the 1st and 3rd days of life of the neonates. The data were analyzed using univariate and multivariate statistical analysis. A unique metabolic pattern of enhanced metabolites was identified in the NICU-admitted LPs from the 1st day of life. Metabolic profiles were distinct in LPs presenting with respiratory distress syndrome (RDS). The discrepancies likely reflect differences in the gut microbiota, either due to variations in nutrient intake or as a result of medical interventions, such as the administration of antibiotics and other medications. Altered metabolites could potentially serve as biomarkers for identifying critically ill LP neonates or those at high risk for adverse outcomes later in life, including metabolic risks. The discovery of novel biomarkers may uncover potential targets for drug discovery and optimal periods for effective intervention, offering a personalized approach.

## 1. Introduction

Late preterm (LP) neonates are defined as newborns born between 34^0/7^ and 36^6/7^ weeks of gestational age [1]. In the United States, they represent 9.1% of all neonates and nearly three-quarters of preterm neonates [2]. Late preterms are characterized by an increased morbidity and mortality risk compared to term neonates. Associated morbidities include neurodevelopmental impairment, feeding difficulties, hypoglycemia, impaired immunological response, sepsis, intraventricular hemorrhage, periventricular leukomalacia, and respiratory disorders such as respiratory distress syndrome (RDS), transient tachypnea of the newborn, pneumonia, apnea, and pulmonary hypertension [1].

The term “metabolomics” refers to the systematic study of a broad range of small endogenous molecule metabolites (<1500 Da) in a biological sample and is a rapidly expanding field of system biology [3]. Τhe quantification of small molecules, such as peptides, lipids, organic acids, vitamins, amino acids, drugs, and other chemicals, enables the identification of metabolic signatures and alterations associated with health and disease states [4]. Given that changes in the individual metabolic profile occur much earlier than any clinically detectable sign or symptom, metabolomics is crucial in the context of the precision medicine approach [5]. Since the metabolome represents the outcome of biochemical processes regulated by proteins derived from gene expression, it provides the closest relationship with the clinical phenotype. Consequently, the utilization of metabolomics as a tool for the validation of new biomarkers for the early diagnosis or prognosis of pathophysiological conditions is constantly growing.

Comprehensive metabolomic analyses of blood have been performed; however, other approaches, using non-invasive matrices to profile metabolites, are gaining popularity [6,7]. Such approaches include the use of a fecal metabolomic profile to predict disease outcomes since fecal metabolites reflect gut microbial composition. It is well known that early colonization of the infant gut has long-term health implications that involve immune system development, growth, cognitive development, and the onset of childhood diseases such as asthma, allergies, and obesity [8,9,10,11,12,13]. The impact of mode of delivery, antibiotic usage, and early life feeding practices on the infant fecal metabolome has also been described by several studies. Distinct fecal metabolomes have also been detected between formula-fed and breastfed infants [14,15,16,17,18].

In addition, urine metabolomics is attracting particular attention, as urine is easily accessible, readily available, and non-invasively collected [19]. In addition, its ease of collection allows for serial sampling. This is particularly important in the case of neonates, particularly those born preterm, who also have small circulating blood volumes. Another important advantage of urine metabolites is that they are a rich source of cellular metabolites originating from blood filtration in the kidney, which represent the final state of metabolism, while the metabolites in the blood may continue to participate in metabolism [7].

More than 600 medical conditions have been associated with the urine metabolites identified thus far [20,21], including obesity, cancer, infectious diseases, and neurological diseases [22,23,24,25]. Furthermore, physiological conditions such as ovulation, pregnancy, diet, and exercise induce metabolomic changes in the urine [26].

Nuclear magnetic resonance (NMR) spectroscopy and mass spectrometry (MS) are the two dominant analytical techniques used for the metabolomic analysis of biofluids. NMR-based analysis, followed by multivariate analysis of the obtained NMR spectra, offers numerous advantages, including that the data are more easily quantitated and highly reproducible [27].

Little is known about metabolic maturation in neonates and its relationship with neonatal morbidity and short- and long-term health outcomes. Even less is known about the metabolic signatures of preterm neonates and the impact of nutritional or pharmaceutical interventions, hospitalization in the NICU, and medical conditions, such as RDS, on the neonatal metabolic profile. There is also a paucity of data regarding the normal urine concentration values for a large number of metabolites in neonates.

The aim of the present study was to identify possible urine metabolites with differing levels between LP neonates hospitalized in the NICU and healthy LP neonates using NMR metabolomics. Additionally, the study aimed to compare the urine metabolic profiles of LP neonates hospitalized in the NICU with RDS and those of neonates hospitalized in the NICU without RDS to define the metabolic patterns associated with this pathology.

## 2. Materials and Methods

### Study Population

The current study was conducted in the neonatal ward and the neonatal intensive care unit (NICU) of the University General Hospital of Patras and the Department of Pharmacy of the University of Patras. Fifty-one (51) neonates that were admitted to the NICU and, additionally, 23 age-matched healthy LPs admitted to the maternity ward from April 2017 to December 2018 were initially included in the study.

The biological sample obtained from each neonate was urine, which was collected with the aim of monitoring the metabolic adaptations taking place during the first days after birth, specifically on the first and third days of life, and comparing them with the metabolism of normal LP infants. Proton nuclear magnetic resonance (^1^H NMR) spectroscopy was used for urine metabolomic analysis. The data were analyzed by univariate and multivariate statistical analysis. Clinical and NMR data from 18 age-matched healthy LP neonates are part of a previous study conducted by our research group [28].

Due to technical issues, including sample inappropriateness, urine samples were obtained on both the 1st and 3rd days of life from only 31 of the neonates admitted to the NICU and 12 healthy neonates from the maternity ward. The neonates admitted to the NICU were further divided into two groups: (a) neonates who developed respiratory distress syndrome (RDS) (*n* = 17), and (b) neonates who did not develop RDS (*n* = 14). Clinical metadata, including neonatal sex, mode of conception, delivery mode, parity, maternal smoking, maternal diseases, maternal medications, maternal infection, birth weight, length, head circumference, requirements of mechanical ventilation, CPAP or O_2_, neonatal diseases, neonatal sepsis, neonatal medications, feeding information, presence of jaundice, RDS, and premature rupture of membranes, were recorded for each neonate participating in the study.

Comparisons of NMR urine metabolic profiles were conducted between 4 different pairs of neonates for the 1st and 3rd days of life, as follows:A1. 1st day NICU LPs (*n* = 51) and healthy age-matched LPs (controls) (*n* = 23).A2. 3rd day NICU LPs (*n* = 31) and healthy age-matched LPs (*n* = 12; controls).B1. 1st day NICU LPs with RDS (*n* = 17) and 1st day NICU LPs without RDS (*n* = 14).B2. 3rd day NICU LPs with RDS (*n* = 9) and 3rd day NICU LPs without RDS (*n* = 9).C1. 1st day NICU LPs with RDS (*n* = 17) and healthy age-matched LPs (*n* = 21).C2. 3rd day NICU LPs with RDS (*n* = 9) and healthy age-matched LPs (*n* = 12).D1. 1st day NICU LPs without RDS (*n* = 14) and healthy age-matched LPs (*n* = 21).D2. 3rd day NICU LPs without RDS (*n* = 12) and healthy age-matched LPs (*n* = 9).

The study was approved by the Research Ethics Committee of the University General Hospital of Patras (IRB number: 154/16.03.2017) and was in accordance with the Declaration of Helsinki of 1975, as revised in 2013. Written informed consent was obtained from the parents of each participating neonate.

## 3. Sample Preparation and NMR Analysis Methodology

Urine samples were collected from the neonates during the 1st and 3rd days of extrauterine life using adhesive pediatric urine collection bags. Immediately after collection, the samples were transferred to sterile vials, centrifuged at 1500× *g* for 10 min, and the supernatant stored at −80 °C until the analysis.

Urine sample preparation as well as NMR spectra acquisition and processing were conducted following the standard metabolomic procedures and methodology as reported by Georgakopoulou et al., 2020, and Georgiopoulou et al., 2022 [28,29].

## 4. Statistical Analysis

The statistical analysis of all the NMR metabolomic data conducted in this study follows the methodology that was previously described by Georgakopoulou et al., 2020; Chasapi et al., 2022; and Matzarapi et al., 2022 [28,30,31]. Multivariate and univariate analysis of urine NMR data was performed using SIMCA 16.0.1 (Umetrics, Umeå Sweden), MetaboAnalyst 5.0, and RStudio 3.5.2 [32]. The multivariate methods principal component analysis (PCA) and partial least squares—discriminant analysis (PLS-DA) were applied to urine NMR data, which were autoscaled before analysis. The quality of the PLS-DA and OPLS-DA models is described by the R2 and Q2 parameters. The cumulative values of R2 and Q2 were calculated via 7-fold cross-validation, and a permutation test with 200 random arrangements of y-variables evaluated the statistical significance of the PLS-DA model. Autoscaling was selected as the most suitable scaling method for the current dataset. A univariate analysis of the successfully assigned metabolites was conducted. Urinary metabolites with no overlapping proton signals were separately integrated, and their levels were analyzed. The analysis was completed via the non-parametric Kruskal test, and for RDS day-group comparison (level of significance, a = 0.01), a paired *t*-test was used. Statistical significance, reported as *p*-values, was estimated after applying the false discovery rate (FDR) correction [33].

## 5. Results

For the neonates hospitalized in the NICU, the gestational age was 34.93 ± 0.929 weeks (mean ± SD) (min: 33, max: 36), and the maternal age was 32.17 ± 6.28 years old (min: 22, max: 49). The Apgar scores at 5 min and 10 min were 7.93 ± 1.15 and 9.17 ± 0.84, respectively. Neonatal birth weight was 2286.85 ± 515.51 g (min: 1400, max: 4060), body length was 46.02 ± 2.59 cm (min: 40, max: 53), and head circumference was 31.89 ± 1.87 cm (min: 28, max: 36.5). The majority of the neonates (86.40%) were born by cesarean section.

Intrauterine growth retardation was documented in 22.60% of the neonates. Respiratory distress was recorded in 54.80% of the NICU neonates who required mechanical ventilation, CPAP, or oxygen. Mechanical ventilation was required for 1.92 ± 1.13 days (min: 1, max: 4), CPAP was required for 1.48 ± 0.75 days (min: 1, max: 4) and oxygen was required for 1.41 ± 0.64 days (min: 1, max: 3). Sixteen neonates presented with early-onset sepsis, eight had late-onset sepsis, and 5.55% had pneumothorax. Jaundice was observed in 38.90% of the neonates, and five neonates developed hypocalcemia, whereas 96.20% of them exhibited no metabolic diseases. No neonatal deaths occurred. On the 5th day of life, 94.45% of the neonates were on antibiotic treatment. Additionally, on the 5th day of life, the majority of the neonates were exclusively formula-fed (55.60%), 31.50% were both formula-fed and breastfed, 11% were on fortifiers, and only 1.90% were exclusively breastfed. On the day of discharge from the NICU, 51% of the neonates were exclusively formula-fed, 43% were both formula-fed and breastfed, 2% were on fortifiers, and 2% were exclusively breastfed.

Among the healthy neonates, 45.80% were breastfed, 21.70% required antibiotics, and 58.30% were born by cesarean section.

The majority of the mothers were not smokers; 9.30% developed preeclampsia, 13% developed gestational diabetes, and 11.10% had hypothyroidism. Nearly half of the mothers (51.85%) were administered corticosteroids (betamethasone) during the pregnancy period; 9.26% received thyroxin; 7.41% received antibiotics; and 5.5% received atosiban. Clinical and epidemiological data on the participating neonates and their mothers are presented in Table 1.

### 5.1. NICU LPs vs. Healthy (Age-Matched) LPs (1st and 3rd Days of Life)

The following metabolomic data comprise the first attempt to characterize the differences in the urine metabolic profile between LP neonates admitted into the NICU and healthy, age-matched LPs (controls) during the 1st and 3rd days of life. Specifically, comparisons were made between the ^1^H NMR urine metabolic profiles of 51 NICU and 23 healthy neonates collected on the 1st day of life and of 31 NICU and 12 healthy neonates collected on the 3rd day of life.

An unsupervised PCA model was first applied to obtain an overview of urine metabolic profiles for the 1st and 3rd days of life. The sample distribution indicates a clustering of the NICU neonates’ urine metabolomes already from the 1st day of their lives. The supervised PLS-DA successfully discriminates neonatal urine metabolomes into two classes (Figure 1, Figure 2, Figure 3 and Figure 4). The presence of gluconate (gluconic acid) in NICU neonates’ urine is the most characteristic difference between the examined groups, and its statistical significance is further confirmed by univariate analysis (Table 2, Figure 5 and Figure 6). A unique metabolic pattern of enhanced metabolites was identified in the NICU admitted LPs. Additionally, gluconate and glycolate were increased in the NICU LPs compared to the controls, while acetoacetate (acetoacetic acid), hippuric acid (hippurate), and lactose were significantly decreased in the NICU LPs (Table 2).

### 5.2. NICU LPs with RDS vs. NICU LPs without RDS (Control NICU)

The urine metabolic profiles of LPs hospitalized in the NICU and diagnosed with RDS were analyzed and compared to the urine metabolic profiles of age-matched LPs hospitalized in the NICU but without RDS. PCA analysis of the 1st day urine metabolome (Figure 7) and PLS-DA (Figure 8) were able to discriminate adequately between the two groups with R2Y (cum) = 93.4% and Q2 (cum) = 36.7%, despite the limited population. PCA and PLS-DA multivariate analyses based on the metabolic profiles from urine collected on the 3rd day of newborns’ lives exhibit the same high classification rates (Figure 9 and Figure 10), with R2Y (cum) = 99.5% and Q2 (cum) = 8.37%.

A heatmap visualization of the targeted urine metabolites on the first and third days of life is shown in Appendix A.

Table 3 shows the urine metabolites exhibiting statistically significant differences between the two groups. The involved metabolic pathways based on the discriminant urine metabolite levels are shown in Table 4 and Figure 11.

### 5.3. NICU LPs with RDS vs. Healthy LP Neonates

Urine metabolic profiles of LPs hospitalized in the NICU diagnosed with RDS were analyzed and compared to the urinary metabolic profiles of healthy, age-matched LPs. PCA and PLS-DA models (Figure 12, Figure 13, Figure 14 and Figure 15) lead to a distinct categorization with R2Y (cum) = 99.5% and Q2 (cum) = 91.3% for the 1st day of life and R2Y (cum) = 99.9% and Q2 (cum) = 98.8% for the 3rd day of life.

A heatmap visualization of the targeted urine metabolites on the first and third days of life is shown in Appendix A, respectively.

Table 5 shows the urine metabolites exhibiting statistically significant differences between the two groups. The metabolic pathways based on the discriminant urine metabolite levels involved are shown in Figure 16.

### 5.4. NICU LPs without RDS (Control NICU) vs. Healthy (Age-Matched) LPs

Urine metabolic profiles of LPs hospitalized in the NICU without RDS were analyzed and compared to the urinary metabolic profiles of healthy, age-matched LPs on the 1st and 3rd days after birth. PCA and PLS-DA models (Figure 17, Figure 18, Figure 19 and Figure 20) lead to a distinct categorization based on gluconate, glucose, and lactose.

The metabolic pathways that were predominantly upregulated in the NICU RDS neonates compared to the NICU non-RDS neonates are: taurine-hypotaurine pathway; pyruvate metabolic pathway; inositol phosphate pathway; arginine-proline pathway; phenylalanine, tyrosine, and tryptophan biosynthesis; and glycine, serine, and threonine metabolic pathway.

The metabolic pathways that were triggered in the NICU LP RDS neonates compared to healthy LP neonates are summarized in Table 6 and Figure 16: the metabolic pathway of synthesis and degradation of ketone bodies; the pentose phosphate pathway; the pyruvate metabolic pathway; the pathway of tyrosine metabolism; alanine, aspartate, and glutamate metabolism; arginine and proline metabolism; inositol phosphate metabolism; phenylalanine, tyrosine and tryptophan biosynthesis; and taurine and hypotaurine metabolism.

## 6. Discussion

The current study aspired to determine whether LP neonates hospitalized in the NICU, particularly those with RDS, present a different urine metabolic fingerprint compared to healthy LP neonates or LP neonates hospitalized in the NICU without RDS. The metabolomic analysis, based on information extracted from the urine NMR spectrum at two time points, supports the presence of distinct metabolic profiles between the studied categories from the first day of life. According to our findings, multiple metabolite levels were disturbed in several metabolic pathways and identified as potential biomarkers for NICU hospitalization or neonatal RDS. Considering that metabolic immaturity early in life is associated with poorer growth outcomes and metabolic disease in the future [4], the metabolic discrepancies presented in this study may hold clinical significance.

It has been reported that the size and chemical diversity of the measurable metabolome in healthy neonates are smaller and simpler compared to those of children and adults due to differences in diet, metabolism, and gut microbiota composition [4]. Higher concentrations of essential amino acids, collagen-associated amino acids, and acylcarnitines in neonatal urine may reflect increased neonatal needs due to rapid cell growth and cell division. It is also well known that metabolic fingerprints and biochemical pathways change in a time-dependent manner in newborns [4,28,29]. Diet has been recognized as a key environmental factor that modulates the metabolic function of the gut microbiota. The composition of milk, which is the first food introduced into the gastrointestinal tract, has a direct impact on gut microbiota and infant neurodevelopment [34,35]. This is due to the provision of essential nutrients for bacterial proliferation (i.e., carbohydrates, proteins, iron, human milk oligosaccharides (HMOs), etc.), immunomodulatory molecules, probiotics, and microbes [36,37]. Human bacterial colonization begins during fetal life, and maternally derived microbial metabolites transmitted to infants via human milk influence the neonatal microbiome and may potentially impact infant health [38,39,40]. There seems to be a window of opportunity during early infancy that is fundamentally influenced by the type of feeding and associated with a healthy microbiota profile [41].

It has also been described that different feeding practices, such as breastfeeding, formula-feeding, or mixed feeding, alter infants’ metabolic signatures [36,42]. There is evidence to suggest that the microbiome plays a more significant role in metabolic activity during early life than in later life [43]. The regulation of infant metabolism and gut immune system function by nutritional strategies during the early stages of infant development is associated with the development of metabolic syndromes later in life, including obesity, insulin resistance, type 2 diabetes mellitus, hypertension, cardiovascular diseases, and gastrointestinal diseases [44,45,46,47]. These dysbiotic changes result in reduced microbial diversity [48].

Furthermore, prematurity has been associated with impaired amino acid, carbohydrate, and fatty acid metabolism, followed by an increase in tricarboxylic acid (TCA) cycle metabolites and urinary choline metabolites. Oxidative stress markers are also higher in preterm infants and are further impaired by parenteral nutrition and formula feeding [49]. Exposure to medical treatment may also drive urine metabolomics patterns in hospitalized neonates.

The present study demonstrated that urine metabolite concentrations, such as gluconate and glycolate, were higher in LP neonates admitted to the NICU compared to healthy LPs, whereas others, i.e., acetoacetate, hippurate, and lactose, were lower. Gluconate can be obtained through the direct oxidation of glucose and is degraded by gluconokinase to generate 6-phosphogluconate, which has a crucial physiological role [45,46]. Since gluconate and glycolate are markers of carbohydrate oxidation, increased concentrations in neonates admitted to the NICU may suggest increased oxidative stress in this population. Increased gluconate concentrations may also be attributed to total parenteral nutrition (TPN), as intravenous calcium administration is a source of exogenous gluconate [50,51].

Hippurate reflects microbial activity in the gut [52] and has been recognized as a metabolomic marker of gut microbiome diversity [53]. Reduced gut microbiome diversity has been associated with disorders such as the metabolic syndrome [47]. Hence, lower levels of hippurate in the neonates admitted to the NICU may suggest altered gut microbiota in these neonates and an increased risk for metabolic syndrome in the future.

Acetoacetate is a weak organic acid produced in the liver under poor metabolism conditions, which result in excessive fatty acid breakdown. Through its partial conversion into acetone, an indispensable source of energy in extrahepatic tissues, the substrates for cholesterol, fatty acid, and complex lipid synthesis—i.e., acetoacetyl-CoA and acetyl-CoA—are provided. During the early postnatal period, acetoacetate and beta-hydroxybutyrate are preferred over glucose as substrates for the synthesis of phospholipids and sphingolipids, which are required for brain growth and myelination. In the lung, acetoacetate serves better than glucose as a precursor for the synthesis of lung phospholipids. The synthesized lipids, particularly dipalmitoyl phosphatidylcholine, are incorporated into surfactant and thus have a potential role in supplying adequate surfactant lipids to maintain lung function during the early days of life. Therefore, the lower levels of acetoacetate during the first day of life in the LP neonates admitted to the NICU compared to the healthy LP neonates may indicate impaired brain growth and myelination, as well as reduced capacity for surfactant production, thus increasing susceptibility to respiratory compromise.

Our results also demonstrated upregulation and downregulation of several metabolic pathways in the neonates who developed RDS, as compared to the neonates who were hospitalized in the NICU but did not develop RDS. The upregulated pathways involved arginine and proline metabolism, taurine and hypotaurine metabolism, phenylalanine, tyrosine, and tryptophan biosynthesis, glycine, serine, and threonine metabolism, suggesting impaired glycogenesis and impaired amino acid and protein metabolism in these neonates. It is well known that the concentrations of most of the above amino acids are much higher in newborns compared to adults, and these age-dependent changes have been attributed to changes in cell growth, tissue growth, and muscle metabolism [54]. High levels of urine glycine, serine, and proline in neonates are possibly correlated with increased levels throughout the whole body in order to support collagen synthesis, rapid cell division, and cell growth [55]. Furthermore, hypotaurine is a sulfinic acid with anti-oxidative and cytoprotective actions. It is converted to taurine, one of the most abundant amino acids in humans, with physical roles that include osmotic pressure control, neuromodulation, immunomodulation, and antioxidants. Taurine is also associated with brain development abnormalities. It is involved in the synthesis of bile acids and is received through human milk or infant formula [56]. It has also been suggested that the hypotaurine-taurine system serves as an energy-saving antioxidative mechanism [57].

The downregulation of metabolic pathways in neonates with RDS is suggestive of impaired protein/amino acid, nucleotide, and carbohydrate metabolism. The pathways involved include D-glutamine and D-glutamate metabolism, phenylalanine metabolism, valine, leucine, and isoleucine biosynthesis, purine metabolism, glyoxylate and dicarboxylate metabolism, glutathione metabolism, pentose phosphate pathway, porphyrin and chlorophyll metabolism, valine, leucine, and isoleucine degradation, and nicotinate and nicotinamide metabolism. Glycine and glutamate are important glycogenic amino acids, whereas glycine and glutamine are used for skeletal muscle growth [58]. Glutamate has also been positively correlated with increased BMI and fasting triglycerides, both of which are associated with insulin resistance [59]. Phenylalanine is an essential amino acid utilized for protein synthesis and conversion to tyrosine, a precursor for catecholamine synthesis and a substrate for thyroid hormone and melanin synthesis [60].

Moreover, valine, leucine, and isoleucine are essential, branched-chain amino acids, that are positively correlated with protein consumption [61]. These amino acids are considered to be prognostic for the onset of insulin resistance and type 2 diabetes mellitus [62,63,64,65]. Their metabolism has been implicated in the pathophysiology of not only the metabolic syndrome but also cancer and hepatic disease [66].

Glutathione plays a crucial role in antioxidant defense, nutrient metabolism, and various cellular processes such as gene expression, DNA and protein synthesis, cell proliferation, apoptosis, and the immune response. Glutathione deficiency contributes to oxidative stress, which is involved in chronic disorders including neurological, hematological, respiratory, cardiovascular, and metabolic diseases [67].

Myo-inositol levels were higher on the first day and lower on the third day in neonates with RDS compared to those hospitalized in the NICU without RDS. Myo-inositol is involved in intracellular insulin signaling and has a prominent role in glucose metabolism and transport. It is also a component of structural lipids, such as phosphatidylinositol [68], and cell membranes, thus playing a vital role in cell morphogenesis, lipid synthesis, and cell growth [69]. Metabolic pathways involving myo-inositol are affected by prematurity [70,71,72], which may increase the risk for future cardiometabolic disease [73,74].

In contrast to other metabolites, betaine levels were lower on the first day of life and higher on the third day in neonates with RDS compared to the control NICU neonates. Betaine is an important precursor for acetylcholine, a neurotransmitter, and phospholipid, which is a structural and signaling component of the cell membrane. Low levels of betaine and choline have been associated with neurodevelopmental delays in animals [75]. The increase in urine betaine levels observed on the third day of life in neonates with RDS may suggest an adaptive and protective mechanism against neurodevelopmental disorders.

Neonates hospitalized in the NICU with RDS also exhibited reduced urinary trimethylamine-N-oxide (TMAO) compared to those hospitalized in the NICU without RDS. This may be attributed to varying feeding practices between the two groups, as increased TMAO concentrations have been linked to milk consumption and/or gut microbiota changes due to breastfeeding [58].

Additionally, urine 3-hydroxyisovalerate (3-HVA), a byproduct of the leucine degradation pathway, was lower in neonates with RDS compared to those without RDS on both the first and third days of life. Increased urine 3-HVA has been reported in preterm infants and is associated with gut microbiome alterations and muscle protein turnover [75].

As to the urine metabolome of late preterm neonates with RDS and that of healthy late preterms, the predominant discrepancy involved the synthesis and degradation of ketone bodies. Ketone bodies are an alternative and glucose-sparing fuel source in the absence of a carbohydrate source. This is particularly important during the first hours of life, when glycogen stores are depleted, and protein catabolism contributes little to energy requirements [76]. Neonates with RDS had reduced acetoacetate levels despite their increased energy requirements, which may indicate reduced or slow fatty acid β-oxidation. Acetoacetate is a product of ketone bodies and fatty acid metabolism in liver mitochondria [77], so decreased levels of acetoacetate suggest reduced activity of the ketogenesis pathway or mitochondrial dysfunction.

In addition, neonates with RDS also indicated upregulated pathways involved in amino acid metabolism as well as pyruvate metabolism, which is an intermediate compound in the metabolism of fats, proteins, and carbohydrates. Differences in the levels of TCA-cycle intermediates (oxoglutarate, citrate, and fumarate) were also observed. Furthermore, reduced allantoin levels in neonates with RDS may suggest impaired glomerular filtration rate, as urinary allantoin concentration is thought to accurately reflect glomerular filtration rate since allantoin is not reabsorbed in the proximal tubule [78].

The higher concentrations of pyruvate in the neonates with RDS on the first day of life may also reflect a higher reliance on anaerobic energy production, which is reversed on the third day of life [79,80].

Hypoxanthine is a purine derivative that represents the final product of adenosine triphosphate (ATP) breakdown in muscle. On the first day of life, neonates with RDS had lower levels of hypoxanthine compared to neonates hospitalized in the NICU without RDS, but higher levels in their urine on the third day of life. High levels of hypoxanthine indicate a high rate of ATP turnover [80,81], while decreased levels may suggest an increased ability of the body to conserve or restore the purine nucleotide pool [82]. The progressive weakening of this ability in neonates with RDS may suggest a reduced capacity for energy production and utilization.

In this study, we attempted to identify the characteristic metabolic alterations and pathways associated with late preterm neonates. Our NMR-based metabolomic analysis using urine as a biological matrix is the first, to our knowledge, to provide data on metabolic changes in the urine samples of late preterm neonates admitted to the NICU, with a focus on LP neonates with RDS. The current study demonstrates that the urine metabolic profiles were different between LP neonates hospitalized in the NICU and LP neonates not hospitalized in the NICU, as well as between neonates with RDS and those without RDS, either hospitalized in the NICU or not. The identification of distinct metabolic patterns from the first day of life in neonates with RDS suggests that metabolic alterations occur early in these neonates. The changes observed in metabolic pathways between the first and third days of life may reflect the metabolic adaptation to energy requirements and systemic stress secondary to RDS or other medical conditions requiring hospitalization. The observed metabolic differences may also arise from differences in the gut microbiota due to the differential nutrition between the different groups of neonates or to medical interventions, such as the administration of antibiotics and other medications.

The neonates admitted to the NICU were predominantly formula-fed when not on parenteral nutrition, whereas the healthy LP neonates were breastfed in a higher proportion than the NICU neonates. The differences found in the urine metabolic profile between the two neonatal groups are in line with data from the literature favoring the importance of the early introduction of human milk in late preterm neonates. It is well known that breastfeeding has been the mainstay for early nutrition in premature infants since, in addition to its nutritional quality, it also improves neonatal neurological development and immunity [83,84]. However, breast milk cannot always meet the nutritional needs of premature infants as a single source of nutrition, as they are born in a state of nutritional deficit and need to catch up to ensure their optimal growth and development. Furthermore, premature infants fail to reach the recommended intake of a range of nutrients due to a limited capacity for fluid consumption [85,86]. Adding breast milk fortifier, a multi-nutrient additive, has been shown to effectively supplement the needs of preterm infants for various nutrients [87,88].

Furthermore, donated breast milk, which is pasteurized and stored in milk banks, has gained popularity as a substitute for breastfeeding [89]. Although its bioactive factors are reduced over time, it is considered superior to formula in many aspects [90]. It retains much of the nutrients of fresh milk, such as protein and lipid content that promote physical development [91], oligosaccharide content that regulates immune response, and bioactive substances that reduce necrotizing enterocolitis (NEC), oxidative stress, and sepsis [92]. Donated breast milk is, however, inferior to formula in terms of weight gain and body length growth in premature infants, which can be overcome by adding a fortifier to donated milk [89].

Therefore, we highlight the importance of the early introduction of human milk in neonatal nutrition in reducing metabolic disturbances in late preterm neonates. Health care policies should aim at promoting the adoption of lactation support practices, such as lactation in the NICU, the provision of motivation and encouragement to the mothers of preterm neonates, integrating human milk banking, or appropriately fortifying human milk with nutritional supplements when needed. In addition, awareness should be raised about the importance of reducing medical interventions, such as the use of antibiotics or other medications, to those that are absolutely necessary according to evidence-based protocols. The implementation of antibiotic stewardship strategies in the neonatal population may allow the reduction in unnecessary antibiotic use and the length of exposure to antibiotics through antibiotic justification and the use of standardized durations of therapy.

Overall, these findings may have important implications for the diagnosis, management, and prediction of outcomes in late preterm neonates with RDS. The identification of altered metabolites and pathways in late preterm neonates with RDS can provide valuable insights into the underlying mechanisms of the disease and can also be used as biomarkers for risk assessment and early diagnosis [93,94]. In addition to facilitating diagnosis, the discovery of novel biomarkers may offer windows of opportunity for intervention and reveal potential targets for drug discovery, opening new avenues for personalized medicine in the neonatal population.

In conclusion, our findings, in conjunction with those of other studies, support the applicability of NMR-based urine metabolomics for further understanding the association between mild prematurity, neonatal diseases, integrated metabolism, nutrition, and medical interventions. Information about the metabolic status of LP neonates may enhance the understanding of neonatal metabolic maturation and the identification of early biomarkers useful for diagnosis, tailored management of neonatal diseases, and early prediction of outcomes.

## Figures and Tables

**Figure 1 metabolites-13-00644-f001:**
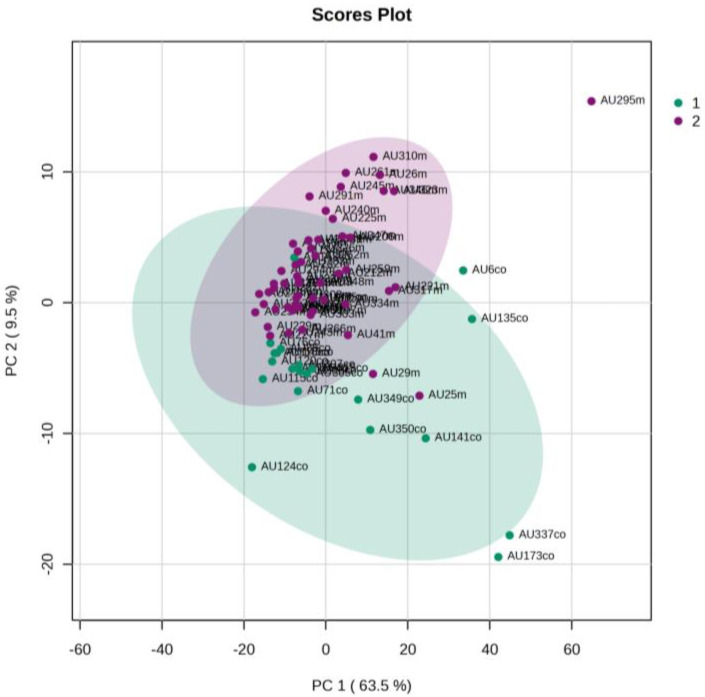
PCA 2D scores plot of 1st day’s urine ^1^H NMR metabolic profiles of NICU late preterms (purple) and healthy late preterms (cyan).

**Figure 2 metabolites-13-00644-f002:**
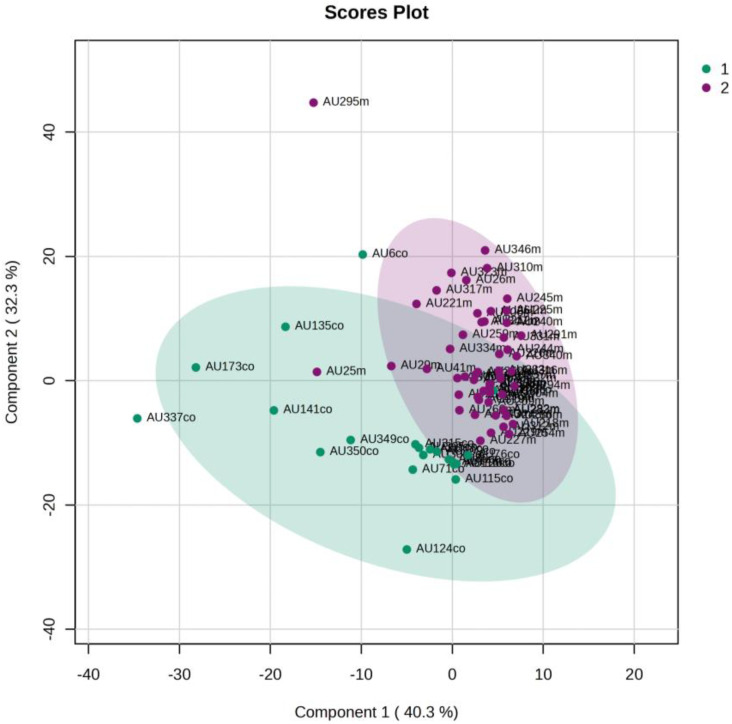
PLS-DA 2D scores plot of 1st day’s urine ^1^H NMR metabolic profiles of NICU late preterms (purple) and healthy late preterms (cyan). R^2^Y (cum) = 83.5%, and Q^2^ (cum) = 66.1%.

**Figure 3 metabolites-13-00644-f003:**
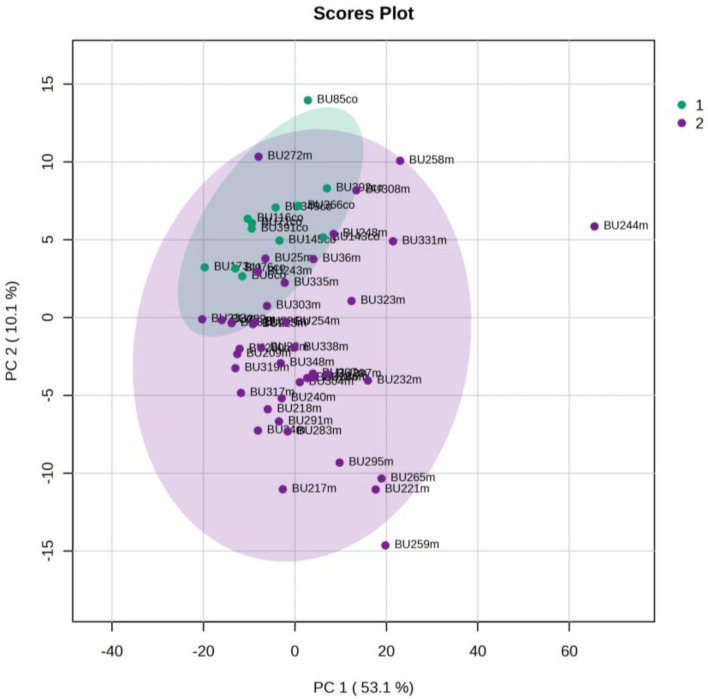
PCA 2D score plot of 3rd day’s urine ^1^H NMR metabolic profiles of NICU late preterms (purple) and healthy late preterms (cyan).

**Figure 4 metabolites-13-00644-f004:**
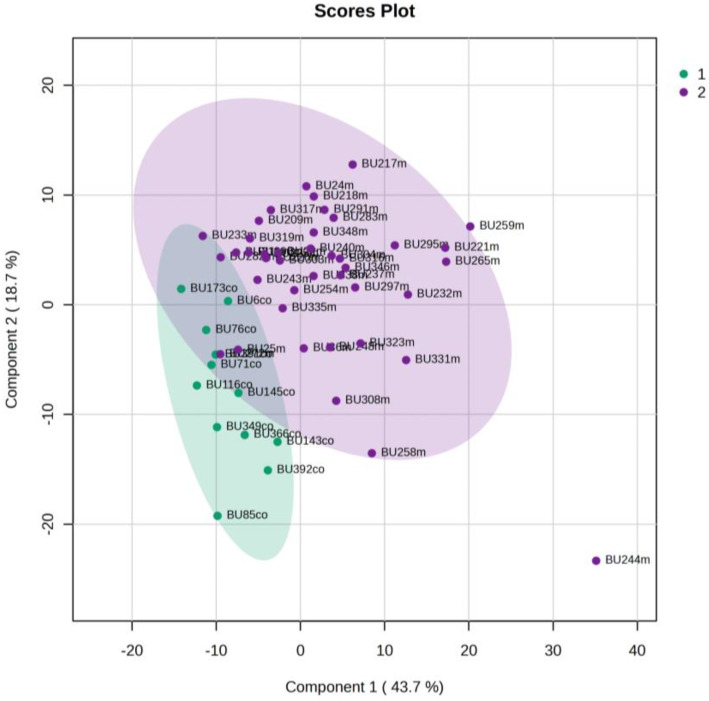
PLS-DA 2D score plot of 3rd day’s urine ^1^H NMR metabolic profiles of NICU late preterms (purple) and healthy late preterms (cyan). R^2^Y (cum) = 92.8% and Q^2^ (cum) = 74.8%.

**Figure 5 metabolites-13-00644-f005:**
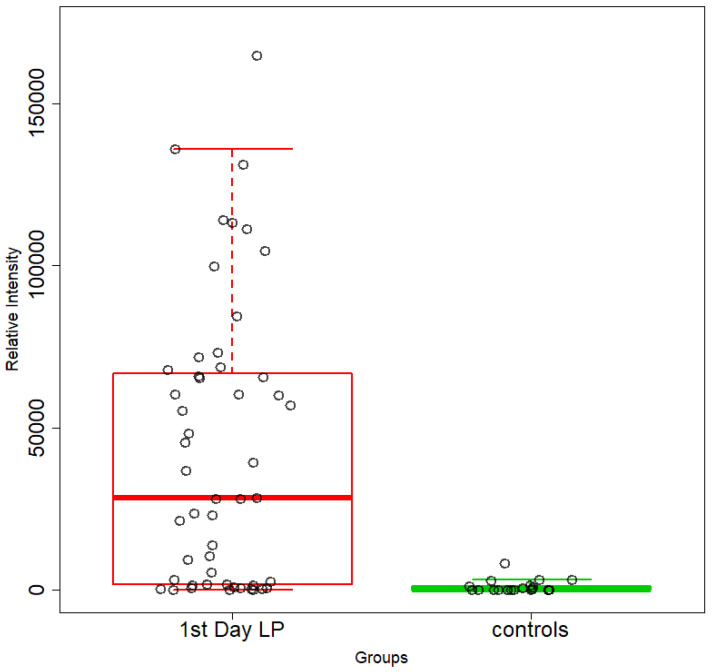
Box plots depicting changes in the relative intensities of gluconate in NICU neonates (red) and healthy LPs (green). FDR *p*-value = 2.463 × 10^−6^ and 1.207 × 10^−5^ on the first day of life.

**Figure 6 metabolites-13-00644-f006:**
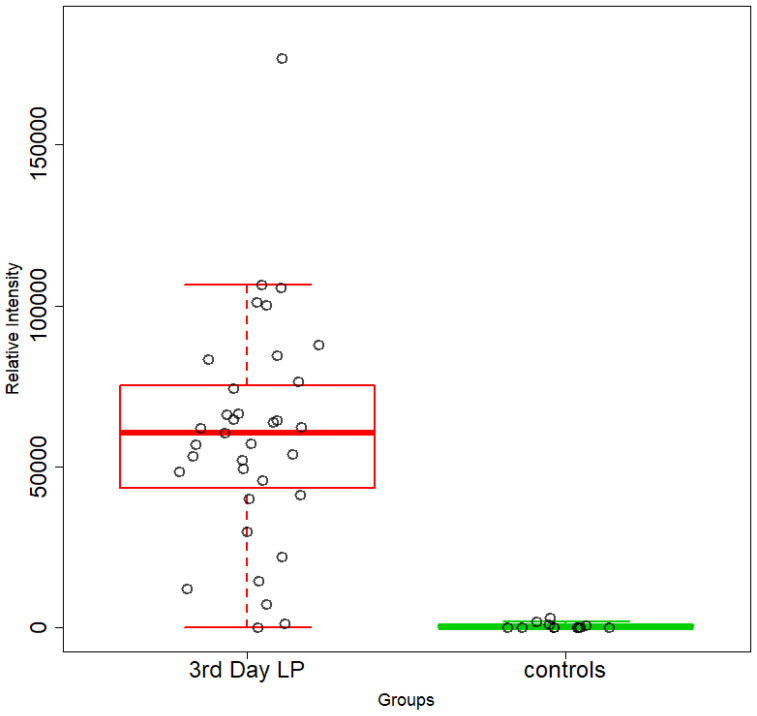
Box plots depicting changes in the relative intensities of gluconate in NICU neonates (red) and healthy LPs (green). FDR *p* value = 2.463 × 10^−6^ and 1.207 × 10^−5^ in the 3rd day of life.

**Figure 7 metabolites-13-00644-f007:**
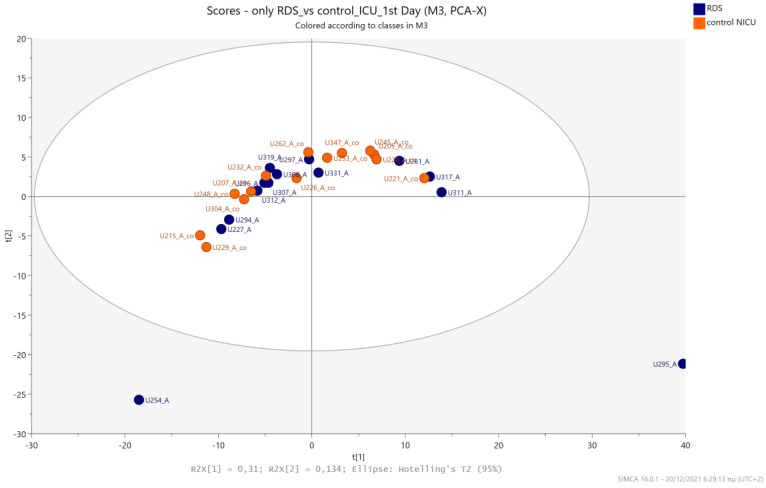
PCA 2D scores plot of 1st day’s urine ^1^H NMR metabolic profiles of NICU LPs with RDS (blue) and control NICU LPs (orange).

**Figure 8 metabolites-13-00644-f008:**
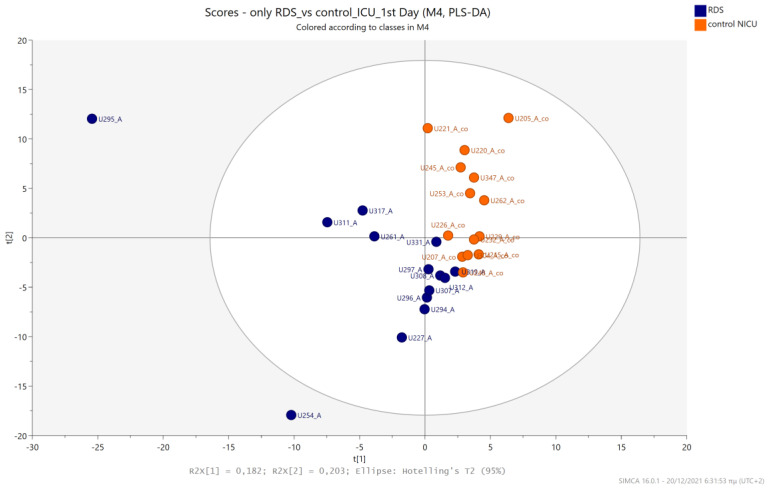
PLS-DA 2D score plot of 1st day’s urine ^1^H NMR metabolic profiles of NICU LPs with RDS (blue) and control NICU LPs (orange). R^2^Y (cum) = 93.4% and Q^2^ (cum) = 36.7%.

**Figure 9 metabolites-13-00644-f009:**
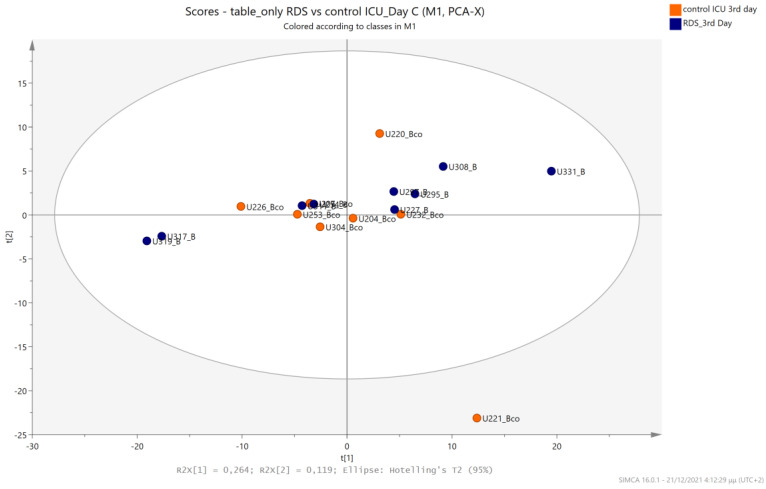
PCA 2D scores plot of 3rd day’s urine ^1^H NMR metabolic profiles of NICU LPs with RDS (blue) and control NICU LPs (orange).

**Figure 10 metabolites-13-00644-f010:**
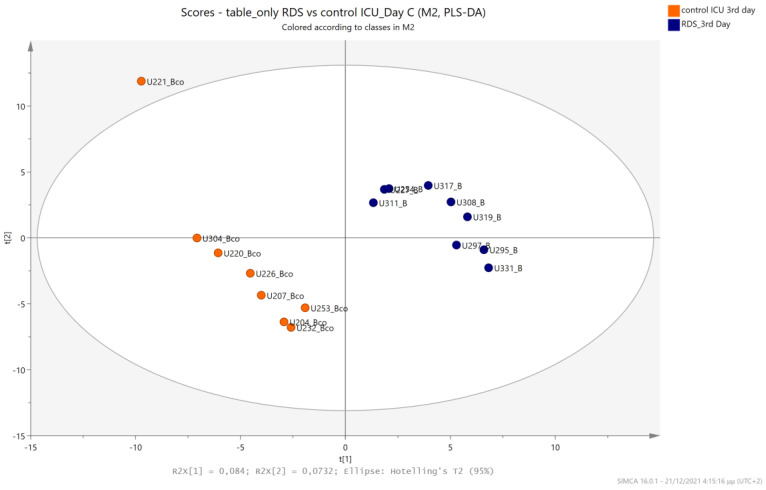
PLS-DA 2D scores plot of 3rd day’s urine ^1^H NMR metabolic profiles of NICU LPs with RDS (blue) and control NICU LPs (orange). R^2^Y (cum) = 99.5% and Q^2^ (cum) = 8.37% (on the 4th component).

**Figure 11 metabolites-13-00644-f011:**
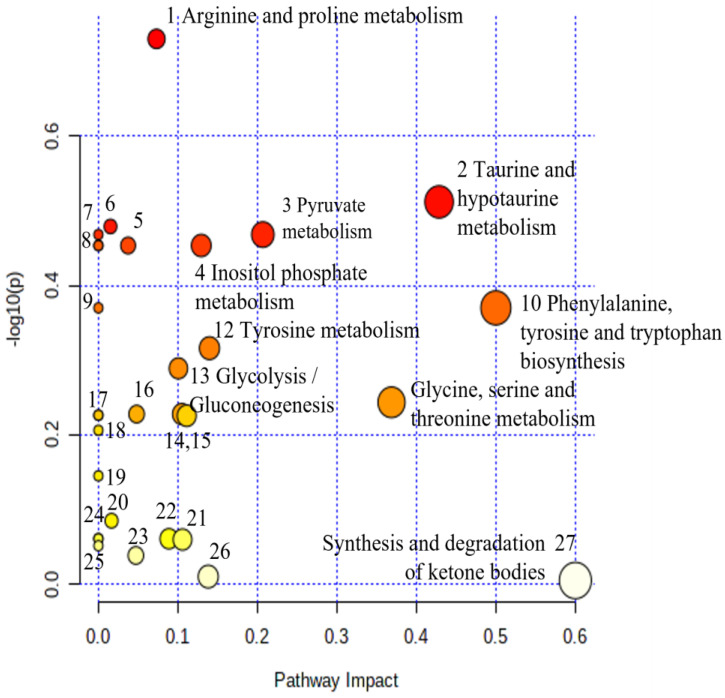
Pathway analysis of the urine of ΝICU late preterms’ urine with RDS versus control NICU late preterms metabolic data displaying metabolic pathways arranged by scores from pathway enrichment analysis (y axis) and from topology analysis (x axis). Each point represents a metabolic pathway. The color and size of each circle are based on *p*-values and pathway impact values, respectively. 5: Phosphatidylinositol signaling system; 6: Primary bile acid biosynthesis; 7: Cysteine and methionine metabolism; 8: Ascorbate and aldarate metabolism; 9: Ubiquinone and other terpenoid-quinone biosynthesis; 14: Butanoate metabolism; 15: Citrate cycle (TCA cycle); 16: Alanine, aspartate, and glutamate metabolism; 17: D-Glutamine and D-glutamate metabolism; 18: Phenylalanine metabolism; 19: Valine, leucine, and isoleucine biosynthesis; 20: Purine metabolism; 21: Glyoxylate and dicarboxylate metabolism; 22: Glutathione metabolism; 23: Pentose phosphate pathway; 24: Porphyrin and chlorophyll metabolism; 25: Valine, leucine, and isoleucine degradation; 26: Nicotinate and nicotinamide metabolism.

**Figure 12 metabolites-13-00644-f012:**
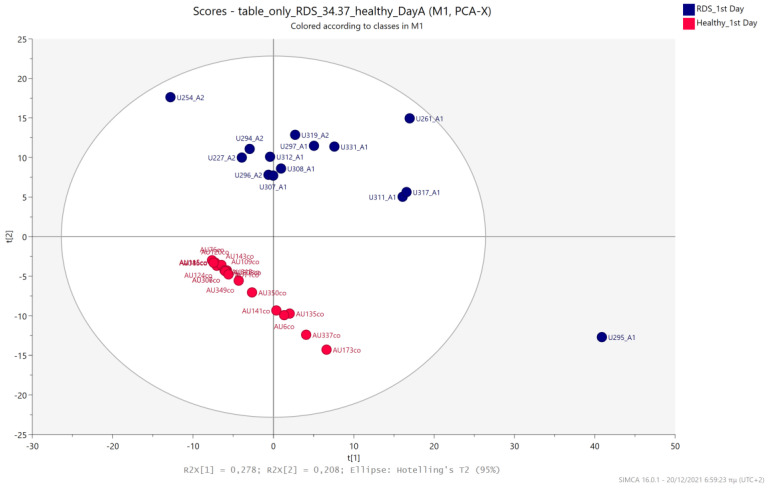
PCA 2D scores plot of 1st day’s urine ^1^H NMR metabolic profiles of NICU LPs with RDS (blue) and healthy LPs (red).

**Figure 13 metabolites-13-00644-f013:**
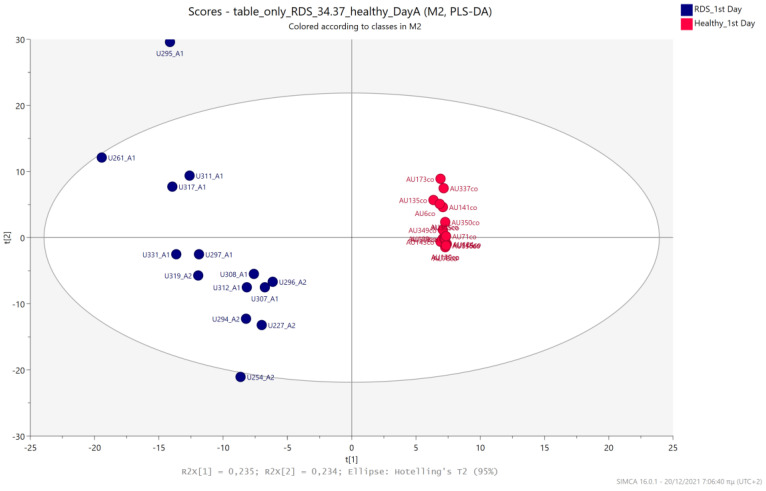
PLS-DA 2D scores plot of 1st day’s urine ^1^H NMR metabolic profiles of NICU LPs with RDS (blue) and healthy LPs (red). R^2^Y (cum) = 99.5% and Q^2^ (cum) = 91.3%.

**Figure 14 metabolites-13-00644-f014:**
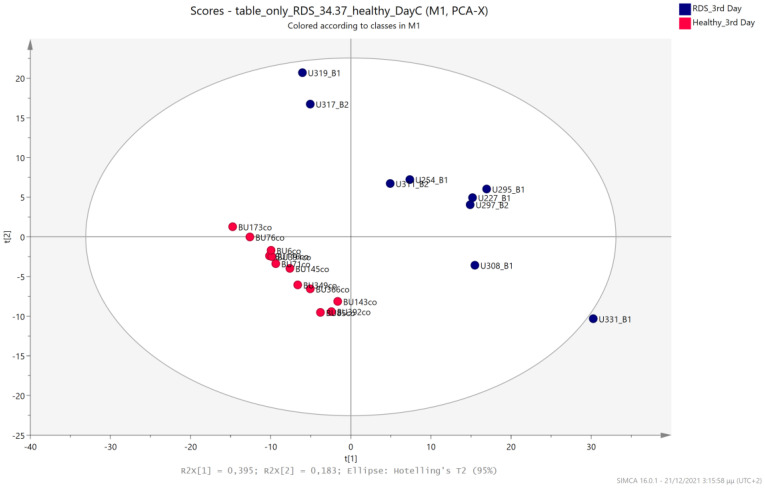
PCA 2D scores plot of 3rd day’s urine ^1^H NMR metabolic profiles of NICU LPs with RDS (blue) and healthy LPs (red).

**Figure 15 metabolites-13-00644-f015:**
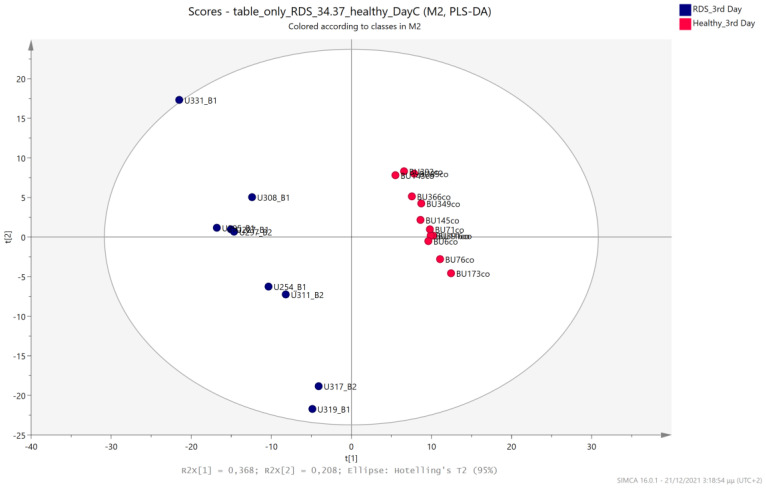
PLS-DA 2D scores plot of 3rd day’s urine ^1^H NMR metabolic profiles of NICU LPs with RDS (blue) and healthy LPs (red). R^2^Y (cum) = 99.9% and Q^2^ (cum) = 98.8%.

**Figure 16 metabolites-13-00644-f016:**
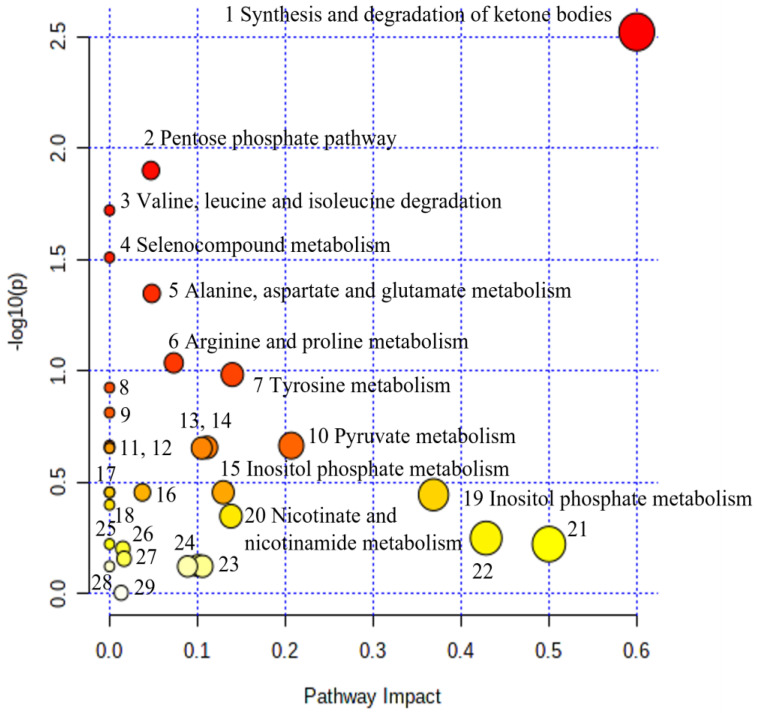
Pathway analysis of the altered metabolites from the late preterms’ urine with RDS versus healthy late preterms’ urine. Metabolic data displaying metabolic pathways arranged by scores from pathway enrichment analysis (y axis) and from topology analysis (x axis). Each point represents a metabolic pathway. The color and size of each circle are based on *p*-values and pathway impact values, respectively. 8: Aminoacyl-tRNA biosynthesis; 9: Phenylalanine metabolism; 11: Cysteine and methionine metabolism; 12: D-Glutamine and D-glutamate metabolism; 13: Citrate cycle; 14: Butanoate metabolism; 16: Phosphatidylinositol signaling system; 17: Ascorbate and aldarate metabolism; 18: Valine, leucine, and isoleucine biosynthesis; 21: Phenylalanine, tyrosine, and tryptophan biosynthesis; 22: Taurine and hypotaurine metabolism; 23: Glyoxylate and dicarboxylate metabolism; 24: Glutathione metabolism; 25: Ubiquinone and other terpenoid-quinone biosynthesis; 26: Primary bile acid biosynthesis; 27: Purine metabolism; 28: Porphyrin metabolism; 29: Glycerophospholipid metabolism.

**Figure 17 metabolites-13-00644-f017:**
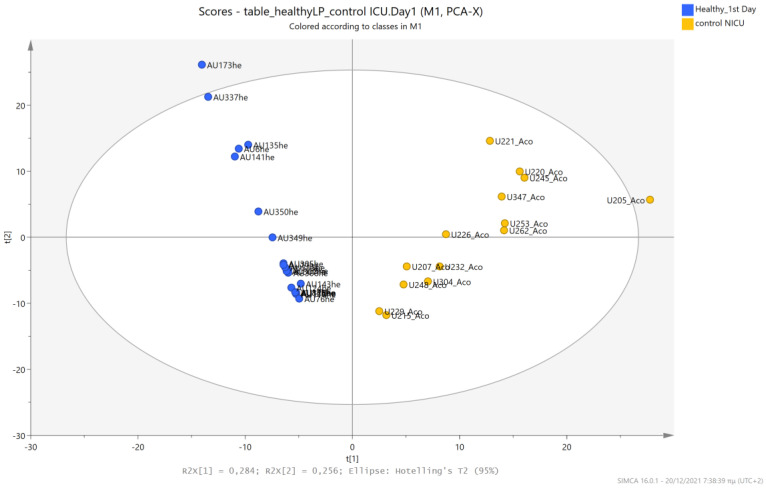
PCA 2D scores plot of 1st day’s urine ^1^H NMR metabolic profiles of control NICU LPs without RDS (yellow) and healthy LPs (cyan).

**Figure 18 metabolites-13-00644-f018:**
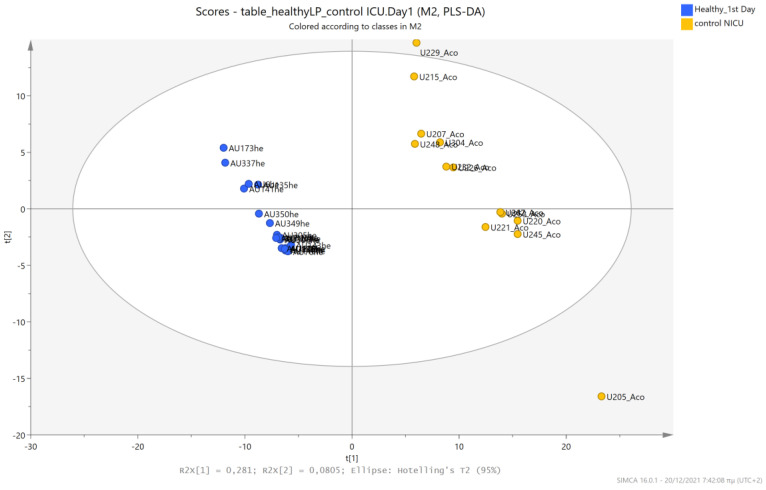
PLS-DA 2D scores plot of 1st day’s urine ^1^H NMR metabolic profiles of control NICU LPs without RDS (yellow) and healthy LPs (cyan). R^2^Y (cum) =99.9% and Q^2^ (cum) = 96.3%.

**Figure 19 metabolites-13-00644-f019:**
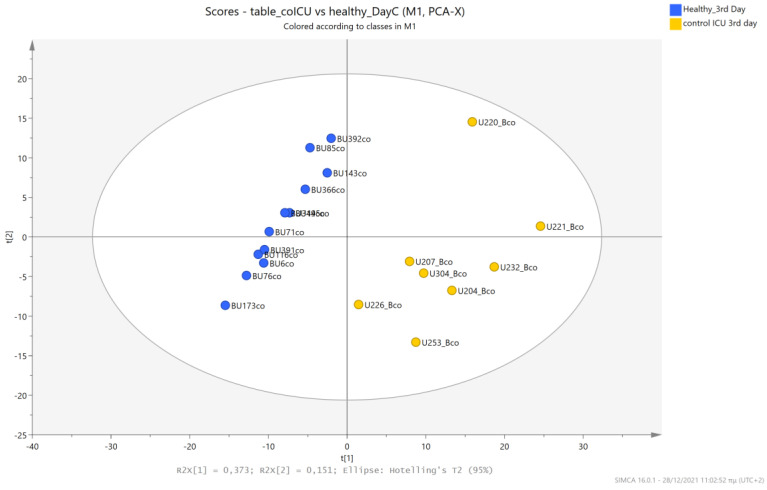
PCA 2D scores plot of 3rd day’s urine ^1^H NMR metabolic profiles of control NICU LPs without RDS (yellow) and healthy LPs (cyan).

**Figure 20 metabolites-13-00644-f020:**
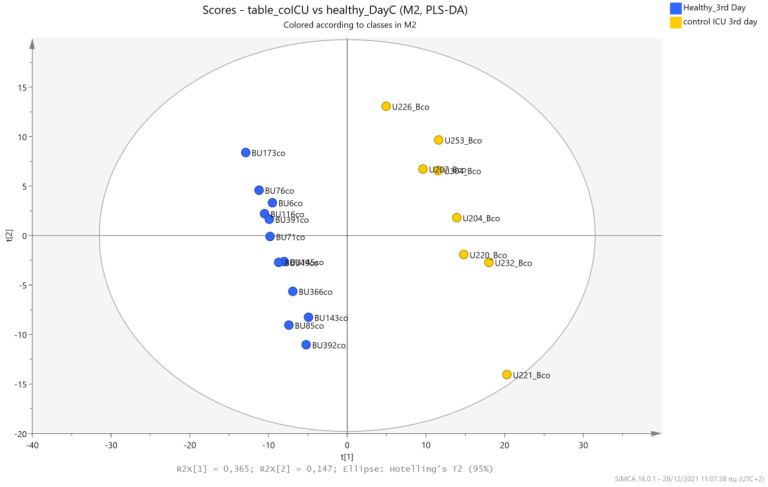
PLS-DA 2D score plot of 3rd day’s urine ^1^H NMR metabolic profiles of control NICU LPs without RDS (yellow) and healthy LPs (cyan). R^2^Y (cum) =99.9% and Q^2^ (cum) = 98.8%.

**Table 1 metabolites-13-00644-t001:** Clinical and epidemiological data of the participating neonates and their mothers.

	*N*	%
**Sex**		
Male	28	51.9
Female	26	48.1
**IVF**		
Yes	7	13.0
No	47	87.0
**Delivery mode**		
Normal	7	13.0
Cesarean	47	87.0
**Parity**		
1st	29	53.7
2nd	18	33.3
3rd	6	11.1
4th	1	1.9
**Maternal smoking**		
Yes	2	3.7
No	52	96.3
**Preeclampsia**		
Yes	5	9.3
No	49	90.7
**Maternal medications**		
None	16	29.63
Betamethasone	28	51.85
Thyroxin	5	9.26
Antibiotics	4	7.41
Atosiban	3	5.55
Thyroxine	2	3.70
Ampicilline-Sulbactam	1	1.85
Amitriptilline	1	1.85
Dexamethasone	1	1.85
Ceftriaxone	1	1.85
Nifedipine	1	1.85
Tinzaparin	1	1.85
Nifedipine	1	1.85
Betamethasone	1	1.85
**Maternal diseases**		
None	35	64.81
Hypothyroidism	6	11.11
Hashimoto	2	3.70
Group B streptococcus colonisation	2	3.70
B-thalasaemia	1	1.85
Migraine	1	1.85
Mild renal failure	1	1.85
Pericarditis	1	1.85
Myopia	1	1.85
Placenta abruption	1	1.85
Uterine fibroids	1	1.85
Oligamnio	1	1.85
Aortic coarctation	1	1.85
**Gestational diabetes**		
No	47	87.0
Diet only	6	11.1
Insulin	1	1.9
**IUGR**		
Yes	12	22.6
No	41	77.4
**Multiple gestation**		
No	38	70.4
Twins	15	27.8
Triplets	1	1.9
**Twin-twin transfusion**		
Yes	0	0.0
No	54	100.0
**Chorioamnionitis**		
Yes	1	1.9
No	53	98.1
**Maternal antibiotics**		
No	2	3.7
Yes < 4 h	39	72.2
Yes	13	24.1
**RDS**		
Yes	17	31.5
No	37	68.5
**PDA**		
Yes	0	0.0
No	54	100.0
**IVH**		
No	52	96.3
Grade1	2	3.7
Grade2	0	0.0
Grade3	0	0.0
Grade4	0	0.0
**Congenital infection**		
Yes	2	3.7
No	52	96.3
**Early-onset sepsis**		
Νο	38	70.4
Yes (- cult)	10	18.5
Yes (+cult,+gram)	4	7.4
Yes (+cult,-gram)	0	0.0
Yes (+cult, fungus)	2	3.7
**Late-onset sepsis**		
Νο	46	85.2
Yes (-culture)	2	3.7
Yes (+culture, +gram)	6	11.1
Yes (+culture, -gram)	0	0.0
Yes (+culture, fungus)	0	0.0
**NEC**		
Yes	1	1.9
No	53	98.1
**Jaundice**		
No	33	61.1
Phototherapy	21	38.9
Exchange transfusion	0	0.0
**Hypocalcemia**		
Yes	5	9.3
No	49	90.7
**Metabolic diseases**		
No	50	96.2
Yes	2	3.8
**Other diseases**		
None	38	70.37
ASD	3	5.55
Pneumothorax	3	5.55
Choroid cysts	2	3.70
Meconium aspiration	1	1.85
Conexingene	1	1.85
Hypospadias	1	1.85
Polycystic kidneys	1	1.85
Hydronephrosis	1	1.85
Laryngomalacia	1	1.85
Congenital heart disease	1	1.85
**Death**		
Yes	0	0.0
No	54	100.0
**Medications DOL 5**		
None	3	5.55
Ampicillin/gentamicin	31	57.41
Ampicillin/cefotaxine	6	11.11
Ampicillin/gentamicin/teicoplanin	4	7.41
Meropenem/vancomycin	3	5.55
Ampicillin/gentamicin/micafungin	1	1.85
Ampicillin/gentamicin/amphotericin	1	1.85
Ampicillin/gentamicin/fluconazole	1	1.85
Vitamin D	1	1.85
Meropenem/vancomycin/rifampicin	1	1.85
Ampicillin/cefotaxime/teicoplanin	1	1.85
Ampicillin/meropenem/vancomycin	1	1.85

IVF: In vitro fertilization; IUGR: Intrauterine growth retardation; RDS: Respiratory distress syndrome; PDA: Patent ductus arteriosus; IVH: Intraventricular haemorrhage; NEC; Necrotic enterocolitis; ASD: Atrial septal defect. DOL: Day of life.

**Table 2 metabolites-13-00644-t002:** The statistically significant metabolites for the discrimination of the NICU late preterms and healthy age-matched late preterms (FDR *p*-value < 0.01) according to the univariate analysis. (s: singlet, d: doublet, m: multiplet).

Day	Metabolites	*δ_H_* (ppm)/Multiplicity	FDR *p*-Values/Effect in NICU LPs	Levels of NICU LPs
**1st**	Acetoacetate	2.25 (s)	0.001	↓
	Gluconate	3.83–3.81 (m)	2.63 × 10^−6^	↑
	Glycolate	3.99 (s)	7.92 × 10^−5^	↑
	Hippurate	7.83–7.81 (d)	4.53 × 10^−6^	↓
	Lactose	4.46–4.43 (d)	4.83 × 10^−5^	↓
**3rd**	Gluconate	3.83–3.81 (m)	1.38 × 10^−5^	↑
	Glycolate	3.99 (s)	0.0003	↑
	Lactose	4.46–4.43 (d)	0.007	↓

**Table 3 metabolites-13-00644-t003:** The statistically significant metabolites for the discrimination of the two groups, late preterm neonates with RDS and control NICU late preterms, according to the univariate analysis.

Day	Metabolites	*p*-Values	Group Levels
1st	1-Methylnicotinamide	0.0013	control NICU > RDS
	Glycine	4.29 × 10^−7^	control NICU > RDS
	Formate	9.66 × 10^−9^	RDS > control NICU
	Alanine	2.80 × 10^−5^	control NICU > RDS
	Hippurate	0.008	control NICU > RDS
	Glucose	1.74 × 10^−5^	control NICU > RDS
	Lactose	0.00017	RDS > control NICU
	4-Hydroxyproline	0.0066	RDS > control NICU
	Gluconate	0.0004	RDS > control NICU
	Dimethylglycine	1.98 × 10^−6^	RDS > control NICU
	Myoinositol	0.0063	RDS > control NICU
	Acetoacetate	2.42 × 10^−6^	RDS > control NICU
	Leucine	3.54 × 10^−7^	control ICU > RDS
	Allantoin	0.00107	RDS > control NICU
	Betaine	2.42 × 10^−6^	control ICU > RDS
	Creatine	0.0006	RDS > control NICU
	Tyrosine	0.007	control ICU > RDS
	4-Hydroxybenzoate	0.0004	control ICU > RDS
	Pyruvate	0.00017	RDS > control NICU
	Dimethylamine	3.54 × 10^−7^	control ICU > RDS
	Trimethylamine	0.006	control ICU > RDS
	Dimethylglycine	2.85 × 10^−5^	RDS > control NICU
	Oxoglutarate	0.00017	control NICU > RDS
	Ethanolamine	0.0006	RDS > control ICU
	TMAO	0.0024	control NICU > RDS
	1-Methylnicotinamide, N1-Methyl-2-pyridone-5-carboxamide	3.60 × 10^−5^	control NICU > RDS
	Hypoxanthine	0.0008	control NICU > RDS
	Trigonelline	0.0015	control NICU > RDS
	4-Hydroxyphenyl acetate	3.64 × 10^−6^	control NICU > RDS
	3-Hydroxyisovaleric acid	0.005	control NICU > RDS
	Indoxyl sulfate	0.0037	RDS > control NICU
3rd	1-Methylnicotinamide	4.96 × 10^−5^	RDS > control NICU
	Glycine	1.04 × 10^−7^	RDS > control NICU
	Formate	0.0011	RDS > control NICU
	Alanine	3.84 × 10^−7^	RDS > control NICU
	Hippurate	2.55 × 10^−5^	control NICU > RDS
	Lactose	0.0058	control NICU > RDS
	4-Hydroxyproline	4.30 × 10^−6^	control NICU > RDS
	Gluconate	6.28 × 10^−5^	RDS > control ICU
	Dimethylglycine	0.0015	control NICU > RDS
	Myoinositol	1.24 × 10^−5^	control NICU > RDS
	Acetoacetate	0.00044	RDS > control NICU
	Leucine	0.0092	RDS > control NICU
	Betaine	1.05 × 10^−6^	RDS > control NICU
	Tyrosine	4.32 × 10^−5^	control NICU > RDS
	4-Hydroxybenzoate	0.00015	control NICU > RDS
	Pyruvate	0.0033	control NICU > RDS
	Dimethylamine	1.05 × 10^−6^	control NICU > RDS
	Trimethylamine	3.84 × 10^−7^	RDS > control ICU
	Dimethylglycine	5.04 × 10^−5^	control NICU > RDS
	Oxoglutarate	7.32 × 10^−5^	control NICU > RDS
	Ethanolamine	5.04 × 10^−5^	control NICU > RDS
	Taurine	0.0011	RDS > control NICU
	1-Methylnicotinamide, N1-Methyl-2-pyridone-5-carboxamide	5.38 × 10^−5^	RDS > control NICU
	Hypoxanthine	1.20 × 10^−5^	RDS > control NICU
	Trigonelline	0.0062	RDS > control NICU
	4-Hydroxyphenyl acetate	2.31 × 10^−5^	control NICU > RDS
	Lys/Arg	3.26 × 10^−6^	control NICU > RDS
	3-Hydroxyisovaleric acid	6.82 × 10^−5^	control NICU > RDS
	Indoxyl sulfate	0.0021	RDS > control NICU

**Table 4 metabolites-13-00644-t004:** Pathway analysis results of the urine metabolite comparison between late preterms with RDS and control NICU late preterms.

No.	Biochemical Pathway	Metabolites Measured	Row *p* Value	Impact
1	Arginine and proline metabolism	Creatine, Hydroxyproline, Pyruvate	1.86 × 10^−1^	0.07
2	Taurine and hypotaurine metabolism	Taurine	3.08 × 10^−1^	0.43
3	Pyruvate metabolism	Pyruvate	3.40 × 10^−1^	0.21
4	Inositol phosphate metabolism	myo-Inositol	3.52 × 10^−1^	0.13
5	Phosphatidylinositol signaling system	myo-Inositol	3.52 × 10^−1^	0.04
6	Primary bile acid biosynthesis	Glycine, Taurine	3.32 × 10^−1^	0.02
7	Cysteine and methionine metabolism	Pyruvate;	3.40 × 10^−1^	0.00
8	Ascorbate and aldarate metabolism	myo-Inositol	3.52 × 10^−1^	0.00
9	Ubiquinone and other terpenoid-quinone biosynthesis	Tyrosine	4.26 × 10^−1^	0.00
10	Phenylalanine, tyrosine, and tryptophan biosynthesis	Tyrosine	4.26 × 10^−1^	0.50
11	Glycine, serine, and threonine metabolism	Betaine, N, N-Dimethylglycine, Glycine, Creatine, Pyruvate	5.71 × 10^−1^	0.37
12	Tyrosine metabolism	Tyrosine, Pyruvate, Acetoacetate, 4-Hydroxyphenylacetate	4.83 × 10^−1^	0.14
13	Glycolysis/Gluconeogenesis	Pyruvate, beta-Glucose	5.14 × 10^−1^	0.10
14	Butanoate metabolism	Acetoacetate, 2-Oxoglutarate	5.95 × 10^−1^	0.11
15	Citrate cycle (TCA cycle)	2-Oxoglutarate, pyruvate	5.92 × 10^−1^	0.10
16	Alanine, aspartate, and glutamate metabolism	2-Oxoglutarate, pyruvate	5.92 × 10^−1^	0.05
17	D-Glutamine and D-glutamate metabolism	2-Oxoglutarate	5.94 × 10^−1^	0.00
18	Phenylalanine metabolism	Hippurate; Tyrosine	6.22 × 10^−1^	0.00
19	Valine, leucine, and isoleucine biosynthesis	Tyrosine	7.16 × 10^−1^	0.00
20	Purine metabolism	Hypoxanthine	8.22 × 10^−1^	0.02
21	Glyoxylate and dicarboxylate metabolism	Glycine, Pyruvate, Formate	8.71 × 10^−1^	0.11
22	Glutathione metabolism	Glycine	8.69 × 10^−1^	0.09
23	Pentose phosphate pathway	Gluconate	9.15 × 10^−1^	0.05
24	Porphyrin metabolism	Glycine	8.69 × 10^−1^	0.00
25	Valine, leucine, and isoleucine degradation	Acetoacetate, Leucine	8.88 × 10^−1^	0.00
26	Nicotinate and nicotinamide metabolism	1-Methylnicotinamide	9.77 × 10^−1^	0.14
27	Synthesis and degradation of ketone bodies	Acetoacetate	9.88 × 10^−1^	0.60

**Table 5 metabolites-13-00644-t005:** The statistically significant metabolites for the discrimination between late preterm neonates with RDS and healthy late preterms according to the univariate analysis.

Day	Metabolites	*p*-Values	Group Levels
**1st**	Alanine	0.047	RDS > healthy
	Lactose	0.016	Healthy > RDS
	Acetoacetate	0.01	Healthy > RDS
	Leucine	0.01	RDS > healthy
	Allantoin	0.015	Healthy > RDS
	4-Hydroxybenzoate	0.019	Healthy > RDS
	Pyruvate	0.05	Healthy > RDS
	Trimethylamine	0.05	Healthy > RDS
	Oxoglutarate	0.033	Healthy > RDS
	Taurine	0.05	Healthy > RDS
	TMAO	0.01	RDS > healthy
	Gluconate	0.01	RDS > healthy
	Trigonelline	0.002	Healthy > RDS
	3-OH-hydroxyisovalerate	2.86 × 10^−6^	RDS > healthy
	Indoxyl sulfate	0.01	RDS > healthy
**3rd**	Alanine	0.004	RDS > healthy
	Gluconate	0.002	RDS > healthy
	Leucine	0.0039	RDS > healthy
	Hypoxanthine	0.0064	RDS > healthy
	Lys/Arg	0.0023	RDS > healthy
	3-OH-hydroxyisovalerate	0.0049	RDS > healthy
	Indoxyl sulfate	0.0023	RDS > healthy

**Table 6 metabolites-13-00644-t006:** Pathway analysis results of the metabolite comparison between late preterms with RDS and healthy late preterms.

No	Biochemical Pathway	Metabolites Measured	Row *p* Value	Impact
1.	Synthesis and degradation of ketone bodies	Acetoacetate	3.00 × 10^−3^	0.60
2.	Pentose phosphate pathway	Gluconate	1.26 × 10^−2^	0.05
3.	Valine, leucine, and isoleucine degradation	Acetoacetate, Leucine	1.90 × 10^−2^	0.00
4.	Selenocompound metabolism	Alanine	3.09 × 10^−2^	0.00
5.	Alanine, aspartate, and glutamate metabolism	Alanine, Pyruvate; 2-Oxoglutarate;	4.48 × 10^−2^	0.05
6.	Arginine and proline metabolism	Creatine, Hydroxyproline, Pyruvate	9.19 × 10^−2^	0.07
7.	Tyrosine metabolism	Tyrosine, Pyruvate; Acetoacetate; 4-Hydroxyphenylacetate;	1.04 × 10^−1^	0.14
8.	Aminoacyl-tRNA biosynthesis	Glycine, Alanine, Leucine, Tyrosine	1.19 × 10^−1^	0.00
9.	Phenylalanine metabolism	Hippurate; Tyrosine	1.54 × 10^−1^	0.00
10.	Pyruvate metabolism	Pyruvate	2.16 × 10^−1^	0.21
11.	Cysteine and methionine metabolism	Pyruvate	2.16 × 10^−1^	0.00
12.	D-Glutamine and D-glutamate metabolism	2-Oxoglutarate	2.22 × 10^−1^	0.00
13.	Citrate cycle (TCA cycle)	2-Oxoglutarate; Pyruvate	2.22 × 10^−1^	0.10
14.	Butanoate metabolism	Acetoacetate, 2-Oxoglutarate	2.20 × 10^−1^	0.11
15.	Inositol phosphate metabolism	myo-Inositol	3.51 × 10^−1^	0.13
16.	Phosphatidylinositol signaling system	myo-Inositol	3.51 × 10^−1^	0.04
17.	Ascorbate and aldarate metabolism	myo-Inositol	3.51e−01	0.00
18.	Valine, leucine, and isoleucine biosynthesis	Leucine	3.99 × 10^−1^	0.00
19.	Glycine, serine, and threonine metabolism	Betaine, N, N-Dimethylglycine, Glycine, Creatine, Pyruvate	3.59 × 10^−1^	0.37
20.	Nicotinate and nicotinamide metabolism	1-Methylnicotinamide, N1-Methyl-2-pyridone-5-carboxamide	4.45 × 10^−1^	0.14
21.	Phenylalanine, tyrosine, and tryptophan biosynthesis	Tyrosine	5.99 × 10^−1^	0.50
22.	Taurine and hypotaurine metabolism	Taurine	5.63 × 10^−1^	0.43
23.	Glyoxylate and dicarboxylate metabolism	Glycine, Pyruvate, Formate	7.53 × 10^−1^	0.11
24.	Glutathione metabolism	Glycine	7.54 × 10^−1^	0.09
25.	Ubiquinone and other terpenoid-quinone biosynthesis	Tyrosine;	5.99 × 10^−1^	0.00
26.	Primary bile acid biosynthesis	Glycine, Taurine	6.28 × 10^−1^	0.02
27.	Purine metabolism	Hypoxanthine	6.96 × 10^−1^	0.02
28.	Porphyrin metabolism	Glycine	7.54 × 10^−1^	0.00
29	Glycerophospholipid metabolism	Ethanolamine	9.90 × 10^−1^	0.01

## Data Availability

The data presented in this study is contained within the article and Appendix A.

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
