# Peer review of "Identification of Novel Biomarkers in Late Preterm Neonates with Respiratory Distress Syndrome (RDS) Using Urinary Metabolomic Analysis"

_metabolites, 2023, doi:10.3390/metabo13050644_

Round 1

Reviewer 1 Report

Dear Authors,

this is a fairly interesting paper that can be published after a few alterations.

It is a very interesting subject. As you highlighted in the manuscript, this study is the first, to knowledge, to provide data on metabolic 413 changes in the urine samples of late preterm neonates admitted to the NICU, with a focus 414 on LP neonates with RDS. Moreover, this study supports the applicability of NMR-based urine metabolomics for understanding further the association between mild prematurity, neonatal diseases, integrated metabolism, nutrition and medical interventions. Thus, any advance in that area is, thus, welcome.

I present some suggestions that I would like you to consider, in order to the article to be published:

Materials and methods:

Are the enrolled patients matched for age and sex? Are they matched for the other icluded variables (such as delivery mode, etc.)?

How long after was metabolomic analysis performed on stored urine samples?

Discussion

Can you suggest some possible early interventions that can be useful in LP patients, such as, for example, food integration, to avoid the metabolomic alterations found?

I have not any comment about English Language

Author Response

We thank the reviewer for all comments, which we believe we have addressed. They were very constructive and have helped us improve our manuscript. The revised manuscript is restructured to comply to the comments made by the reviewer. All revisions in the text are presented in yellow font

Reviewer 1

Dear Authors,

this is a fairly interesting paper that can be published after a few alterations.

It is a very interesting subject. As you highlighted in the manuscript, this study is the first, to knowledge, to provide data on metabolic 413 changes in the urine samples of late preterm neonates admitted to the NICU, with a focus 414 on LP neonates with RDS. Moreover, this study supports the applicability of NMR-based urine metabolomics for understanding further the association between mild prematurity, neonatal diseases, integrated metabolism, nutrition and medical interventions. Thus, any advance in that area is, thus, welcome.

I present some suggestions that I would like you to consider, in order to the article to be published:

Materials and methods:

  1. Are the enrolled patients matched for age and sex? Are they matched for the other included variables (such as delivery mode, etc.)?

Response: We thank the reviewer for the kind comments. The enrolled patients are matched only for age and gestational age. The other variables differ and we believe that the distinct metabolic profiles between the NICU neonates and the healthy neonates are attributed to these differences. We have added some additional information regarding the prevalence of some of the variants in the healthy neonates to the Results section.

  1. How long after was metabolomic analysis performed on stored urine samples?

Response: The authors confirm that the quality of urine samples was maintained throughout the storage and transportation process, which was conducted according to the European Standard Operating Procedures (SOPs) for biobanking and the World Health Organization (WHO) guidelines.

Discussion

Can you suggest some possible early interventions that can be useful in LP patients, such as, for example, food integration, to avoid the metabolomic alterations found?

Response: We have added some suggestions regarding possible early interventions that can be useful in LP neonates in the Discussion section, before the conclusion. We have made reference to early introduction of human milk in the diet of LP neonates, using milk fortifiers and donated milk and by supporting lactation in the NICU. Also, we have highlighted the importance of the implementation of antibiotic stewardship strategies in order to reduce the exposure of LP neonates to antibiotics.

Reviewer 2 Report

Dear authors,

this is an interesting piece of research work, that is well designed, written and presented.  

I have some minor comments for your rerearch:

1) I would suggest you give the clinical implications of this research in more details in the discussion section, and in particular the differences you described between NICU and healthy neonates. How can these differences be helpful in the clinical practice?

2) I also suggest a language editing of the manuscript by a native speaker.

3) In line 176, you mention that 51 NICU neonates... This is probably incorrect as  the study consisted of only 31 NICU neonates. Similarly, in line 177 you mention 35 NICU neonates. This is probably incorrect too.

I suggest a language editing of the manuscript by a native speaker.

Author Response

We thank the reviewer for all comments, which we believe we have addressed. They were very constructive and have helped us improve our manuscript. The revised manuscript is restructured to comply to the comments made by the reviewer. All revisions in the text are presented in yellow font.

Reviewer 2

Dear authors,

this is an interesting piece of research work, that is well designed, written and presented.  

I have some minor comments for your research:

1) I would suggest you give the clinical implications of this research in more details in the discussion section, and in particular the differences you described between NICU and healthy neonates. How can these differences be helpful in the clinical practice?

Response: We thank the reviewer for the kind comments. We have added the clinical implications of our research and how our findings can be helpful in the clinical practice to the Discussion section (last four paragraphs before the conclusion-highlighted in yellow font).

2) I also suggest a language editing of the manuscript by a native speaker.

Respone: The manuscript has been edited by a native speaker. We thank for the comment.

3) In line 176, you mention that 51 NICU neonates... This is probably incorrect as  the study consisted of only 31 NICU neonates. Similarly, in line 177 you mention 35 NICU neonates. This is probably incorrect too.

Response: We apologise for the misunderstanding caused. We have now made clear that the NICU neonates initially enrolled were 51 and the healthy neonates 23, but due to technical reasons and sample inappropriateness, urine samples were obtained on both the 1st and 3rd day of life from only 31 of the neonates admitted to the NICU and 12 healthy neonates from the maternity ward. The numbers have been corrected.

Reviewer 3 Report

The authors attempted to identify the metabolic signature for preterm neonates with Respiratory Distress Syndrome (RDS). To achieve this, they leveraged a cohort of (1) 31 late preterm (LP) neonates admitted to the neonatal intensive care unit (NICU) and (2) 23 healthy LPs with similar ages. They used proton nuclear magnetic resonance (1H NMR) spectroscopy to analyze the metabolomic profile of urine samples from two groups of infants on days 1 and 3 of their life. They analyzed their data using univariate and multivariate statistical analysis, finding a unique metabolic pattern in the NICU-admitted neonates, and differences in the metabolic profile were also identified in neonates with respiratory distress syndrome compared to those without. Although the study highlights the importance of identifying metabolic differences early in life to prevent metabolic disease and could serve as potential biomarkers for identifying critically ill neonates or those at risk of adverse outcomes in later life, a proper comparison with other existing approaches is lacking. Therefore, I would recommend a major revision of the manuscript and I am happy to look at the revised version if they can answer my comments satisfactorily, which I describe in more detail below.

Major issues:

1.     I worry about a lack of comparison with other approaches. For example, it has been shown in the past that there is a potential connection between childhood respiratory diseases and their early-life gut microbiota (Cristina Garcia-Maurino Alcazar et al, the Lancet Microbe 2022). In addition, the connection between gut microbiota and metabolites has been shown in many pieces of literature (Akshit Goyal et al., Nature Communications 2021; Jaeyun Sung et al., Nature Communications 2017). Moreover, I don’t know if using a fecal metabolomic profile to predict disease outcomes is better than using a urine metabolomic profile. Therefore, the authors need to properly summarize other approaches adopted previously and compare their method with previously existing methods if possible.

2.     I would recommend using a machine-learning model such as Random Forest or linear regression to predict the health status of individuals in their later life, based on their metabolomic profiles. There were many computational methods have been developed in the past to predict respiratory disease of infants later in life based on their omics data in their early life (Xuwen Wang, et al, Respiratory Research 2023). In this way, the authors can better prove the predictive power of their metabolomic profiles and test their usefulness in practice. Then based on the trained predictive model, they could once again investigate the importance of each feature and compare the feature importance with the correlation results they presented.

3.     The texts in Figs. 3, 5, and 7 are too small to be visible. Please increase the font sizes.

Minor comments:

1.     Line 16: write the full name of NICU before showing the first acronym.

2.     Line 17: show the full name of NMR, just in case some readers do not know it.

3.     Line 179: please write the full name of PCA “Principal component analysis”.

4.     Line 182: please write the full name of PLS-DA “Partial least squares discriminant analysis”.

5.     For the figure caption of Fig. 6, the “Figure 6: ” is missing before the caption.

Author Response

We thank the reviewer for all comments, which we believe we have addressed. They were very constructive and have helped us improve our manuscript. The revised manuscript is restructured to comply to the comments made by the reviewer. All revisions in the text are presented in yellow font.

The authors attempted to identify the metabolic signature for preterm neonates with Respiratory Distress Syndrome (RDS). To achieve this, they leveraged a cohort of (1) 31 late preterm (LP) neonates admitted to the neonatal intensive care unit (NICU) and (2) 23 healthy LPs with similar ages. They used proton nuclear magnetic resonance (1H NMR) spectroscopy to analyze the metabolomic profile of urine samples from two groups of infants on days 1 and 3 of their life. They analyzed their data using univariate and multivariate statistical analysis, finding a unique metabolic pattern in the NICU-admitted neonates, and differences in the metabolic profile were also identified in neonates with respiratory distress syndrome compared to those without. Although the study highlights the importance of identifying metabolic differences early in life to prevent metabolic disease and could serve as potential biomarkers for identifying critically ill neonates or those at risk of adverse outcomes in later life, a proper comparison with other existing approaches is lacking. Therefore, I would recommend a major revision of the manuscript and I am happy to look at the revised version if they can answer my comments satisfactorily, which I describe in more detail below.

Major issues:

  1. I worry about a lack of comparison with other approaches. For example, it has been shown in the past that there is a potential connection between childhood respiratory diseases and their early-life gut microbiota (Cristina Garcia-Maurino Alcazar et al, the Lancet Microbe 2022). In addition, the connection between gut microbiota and metabolites has been shown in many pieces of literature (Akshit Goyal et al., Nature Communications 2021; Jaeyun Sung et al., Nature Communications 2017). Moreover, I don’t know if using a fecal metabolomic profile to predict disease outcomes is better than using a urine metabolomic profile. Therefore, the authors need to properly summarize other approaches adopted previously and compare their method with previously existing methods if possible.

Response: We thank the reviewer for the kind comments and for the constructive recommendations. We hope that they find our responses satisfactory. The potential connection between early-life gut microbiota and short- or long-term health consequences has been made clearer. Also, reference has been made to the approach of using fecal metabolic profiles to predict disease outcomes and to the advantages of this approach. Although we have mentioned that fecal metabolomics is advantageous due to the ease of collection and the fact that it reflects gut microbiome, a direct comparison between fecal and urine metabolomics has not been made, as literature data are lacking to support the one approach over the other. Also, the purpose of the present study was to highlight the applicability of urine metabolomics in profiling metabolic signatures of late preterm neonates.

  1. I would recommend using a machine-learning model such as Random Forest or linear regression to predict the health status of individuals in their later life, based on their metabolomic profiles. There were many computational methods have been developed in the past to predict respiratory disease of infants later in life based on their omics data in their early life (Xuwen Wang, et al, Respiratory Research 2023). In this way, the authors can better prove the predictive power of their metabolomic profiles and test their usefulness in practice. Then based on the trained predictive model, they could once again investigate the importance of each feature and compare the feature importance with the correlation results they presented.

Response: We would like to thank the reviewer for their suggestions. In the present study, the authors' main interest lies in identifying the differentiated urinary levels and comparing the metabolic profiles using two commonly applied multivariate analysis models, PCA and PLS-DA. This study involves a substantial number of newborns who were enrolled in a clinical trial that commenced in 2016. The pertinent findings from this trial have already been integrated and published in two separate papers by Georgakopoulou et al. (2020) and Georgiopoulou et al. (2022). Therefore, the authors aim to establish a connection between the multivariate analysis methodology of the urinary metabolomic data and the previously published results.

The authors would like to highlight that various machine learning methods and models were tested before concluding and selecting the ones presented in this study. Specifically, the Random Forest model was excluded due to the small sample size, which would not have significantly improved the model's ability to learn new information about the underlying distribution through bootstrap resampling but through shuffling of the cases without drawing several of them.

Moreover, linear regression was not used as the data did not meet the fundamental assumption of a linear relationship between the dependent and independent variables. The features were not normally distributed, and the residuals vs. fits plots of a representative number of significant metabolic features from Hippurate, Gluconate, and Lactose in the following Figures showed that the features were not linear.

In summary, the authors want to emphasize that they conducted a thorough analysis before selecting the machine learning methods and models presented in this study, and each decision was made based on careful consideration of the data.

  1. The texts in Figs. 3, 5, and 7 are too small to be visible. Please increase the font sizes.

Response: The font sizes have been increased. In the current version of the manuscript, Figs 3, 5 and 7 are Figs 7-10, 12-15 and 17-20, respectively.

Minor comments:

  1. Line 16: write the full name of NICU before showing the first acronym.

Response: The full name of NICU has been added to the Abstract where NICU is mentioned for the first time.

  1. Line 17: show the full name of NMR, just in case some readers do not know it.

Response: The full name of NMR, Nuclear Magnetic Resonance, is already provided in line 17.

  1. Line 179: please write the full name of PCA “Principal component analysis”.

Response: The full name of PCA has been added in line 131 where it is first mentioned.

  1. Line 182: please write the full name of PLS-DA “Partial least squares discriminant analysis”.

Response: The full name of PLS-DA has been added in line 132 where it is first mentioned.

  1. For the figure caption of Fig. 6, the “Figure 6: ” is missing before the caption.

Response: We apologise for the oversight. Figure 6 in the previous version of the manuscript is Figure 16 in the current version and “Figure 15” has been added to the caption.

Please, see the non-published material.

Reviewer 4 Report

 The author needs to analyze the relationship between biomarkers in late preterm neonates and newborn gender.

2. This manuscript requires detailed information on how to exclude the impact of other diseases on the analysis of neonates with respiratory distress syndrome in this study.

3. Figure 3 should be relabeled since Figure labels are smear.

4. This study should summarize the functional characteristics of meaningful biomarkers and classify them by source or function, to have implications for the health of newborns.

 The manuscript needs minor editing of English language.

Author Response

We thank the reviewer for all comments, which we believe we have addressed. They were very constructive and have helped us improve our manuscript. The revised manuscript is restructured to comply to the comments made by the reviewer. All revisions in the text are presented in yellow font.

Reviewer 4.

  1. The author needs to analyze the relationship between biomarkers in late preterm neonates and newborn gender.

Response: We thank the reviewer for the comment. The NICU neonates and the healthy neonates enrolled into the study are not matched for sex and an analysis based on sex differences has not been performed. If the reviewer finds it important that such an analysis is included in the manuscript in order to be published, we will have to request an extension from the Editors to perform the analysis.

  1. This manuscript requires detailed information on how to exclude the impact of other diseases on the analysis of neonates with respiratory distress syndrome in this study.

Response: We thank the reviewer for the comment. In order to exclude the impact of other diseases on the analysis of neonates with respiratory distress syndrome we included only neonates with RDS who had no other diseases, such as NEC, sepsis etc. We also compared the NICU neonates with RDS with the non-RDS NICU neonates, who were admitted to the NICU because of other diseases, i. e. NEC, sepsis etc.

  1. Figure 3 should be relabeled since Figure labels are smear.

Response: The reviewer’s comment has been addressed.

  1. This study should summarize the functional characteristics of meaningful biomarkers and classify them by source or function, to have implications for the health of newborns.

Response: The authors have made an attempt to classify the multiple distinct urine metabolites found in the NICU and, particularly, in RDS neonates compared to the healthy neonates, by function. Figures 11 and 16 summarize the metabolic pathways these metabolites are involved in and that are mostly impaired in the studied neonatal groups. Although it is difficult to determine which are the meaningful metabolites and the related metabolic pathways, those participating in carbohydrate, fatty acid, and protein metabolism appear to be particularly impaired. In the Discussion section, an extended reference is made to the physiological roles and functions of meaningful biomarkers, and the processes these metabolites are involved in, i.e. microbial activity, synthesis of cell membrane components, surfactant production, collagen synthesis, cell division and growth, brain growth and myelination, oxidative stress, apoptosis, immune response. Based on the literarure, some of the potential health implications for the neonates are also mentioned, including neurological, hematological, respiratory, hepatic, cardiovascular, metabolic diseases and cancer.

Comments on the Quality of English Language

The manuscript needs minor editing of English language.

Response: The manuscript has been edited by a native English speaker.

Round 2

Reviewer 3 Report

The authors answered our questions. I have no other comments.